# Bioinspired Synthesis of Silver Nanoparticles for the Remediation of Toxic Pollutants and Enhanced Antibacterial Activity

**DOI:** 10.3390/biom13071054

**Published:** 2023-06-29

**Authors:** Sujata Mandal, Sangchul Hwang, Sreekar B. Marpu, Mohammad A. Omary, Victor Prybutok, Sheldon Q. Shi

**Affiliations:** 1Ingram School of Engineering, Texas State University, San Marcos, TX 78666, USA; sujatamandal@txstate.edu; 2Department of Chemistry, University of North Texas, Denton, TX 76207, USA; sreekarbabu.marpu@unt.edu (S.B.M.); mohammad.omary@unt.edu (M.A.O.); 3G. Brint Ryan College of Business, University of North Texas, Denton, TX 76207, USA; victor.prybutok@unt.edu; 4Department of Mechanical Engineering, University of North Texas, Denton, TX 76207, USA

**Keywords:** silver-modified activated carbon, heavy metals, dye, bacteria, adsorption

## Abstract

This research presents a novel and environmentally friendly approach for the synthesis of multifunctional nanobiocomposites for the efficient removal of toxic heavy metal and dye, as well as the disinfection of wastewater microorganisms. The nanobiocomposites (KAC-CS-AgNPs) were prepared by incorporating photochemically generated silver nanoparticles (AgNPs) within a chitosan (CS)-modified, high-surface-area activated carbon derived from kenaf (KAC), using a unique self-activation method. The even distribution of AgNPs was visible in the scanning electron microscopy images and a Fourier transform infra red study demonstrated major absorption peaks. The experimental results revealed that KA-CS-AgNPs exhibited exceptional adsorption efficiency for copper (Cu^2+^), lead (Pb^2+^), and Congo Red dye (CR), and showed potent antibacterial activity against *Staphylococcus aureus* and *Escherichia coli*. The maximum adsorption capacity (mg g^−1^) of KAC-CS-AgNPs was 71.5 for Cu^2+^, 72.3 for Pb^2+^, and 75.9 for CR, and the adsorption phenomena followed on the Freundlich and Langmuir isotherm models and the second-order kinetic model (R^2^ > 0.99). KAC-CS-AgNPs also exhibited excellent reusability of up to four consecutive cycles with minor losses in adsorption ability. The thermodynamic parameters indicated that the adsorption process was spontaneous and endothermic in nature. The bacterial inactivation tests demonstrated that KAC-CS-AgNPs had a strong bactericidal effect on both *E. coli* and *S. aureus*, with MIC calculated for *E. coli* and *S. aureus* as 32 µg mL^−1^ and 44 µg mL^−1^, respectively. The synthesized bioinspired nanocomposite KAC-CS-AgNPs could be an innovative solution for effective and sustainable wastewater treatment and has great potential for commercial applications.

## 1. Introduction

Water pollution resulting from rapid and extensive industrialization, urbanization, and population growth has led to a scarcity of safe water and is a major concern affecting over 1.2 billion people worldwide [1]. Toxic substances, such as heavy metals, dyes, and pathogenic microorganisms, pose a significant threat to human health and ecosystems [2,3,4]. Lead (Pb^2+^) and copper (Cu^2+^) are two common heavy metals found to originate from industrial wastewaters due to mining activities, electrical industry, electroplating, lead pipe construction, and smelting industries. It has been reported that long-term exposure to metallic lead can cause severe damage to the kidney, nerves, brain, liver, as well as the reproductive system [5]. Prolonged ingestion of high doses of copper may result in renal failure, liver damage, and critical poisoning of the human body [6]. The textile industries, printing and dyeing, paper making, rubber, and plastics industries are responsible for generating Congo red (CR), an azo dye [7,8]. The aromatic CR dye possesses a stable structure and easily resists natural degradation and causes significant threat to human health and ecosystems [5]. Therefore, the development of advanced, versatile, environmentally friendly techniques for their removal from water is crucial [9]. Some of the common remediation techniques for the decontamination of effluents from aqueous solutions are adsorption [10], ultrafiltration [11], nanofiltration membranes [12], photocatalytic degradation [13], coagulation/flocculation [14], and biological procedures [15].

Adsorption is an effective technique for the decontamination of effluents from aqueous solutions due to its effectiveness, availability of different adsorbents, simple design, speedy adsorption capacity, low operational cost, easy handling, and effortless regeneration potential [16,17,18,19]. However, the nature and properties of the absorbent are crucial, and activated carbon (AC), which is a popular adsorbent, is expensive and produces toxic sludge [20,21]. There is a need for affordable, abundant, and sustainable materials that are potential sources of AC for the removal of multiple toxic pollutants from water [22]. Porous cellulose and cellulose-based materials, including non-feed industrial energy crops like hemp, coconut, kenaf, agro-industrial crop wastes, straws/stalks, bagasse, and fruit/nut shells, have been applied for heavy metal ions and dye removal [23,24,25,26]. Chitosan (CS) derived from chitin is one of the most abundant polysaccharide biopolymers used extensively for the removal of heavy metals and dyes [27]. Its poor mechanical properties and tiny specific surface area are some of the limitations of CS; hence, the modification of CS to enhance its physicochemical properties is essential [28]. Silver nanoparticles (AgNPs) have shown high bioactivity and are the most encouraging due to their nontoxicity and antimicrobial properties. AgNPs can disrupt pathogen membranes, thus inhibiting bacterial growth by disrupting the enzymatic activities of microorganisms [29].

It should be noted that several modified natural or bioadsorbents, nanoadsorbents, and benign nanocomposites fabricated with AgNPs, are reported in previous studies for the removal of single heavy metal ions [30], or two types of pollutants like heavy metals ions and dye [31] or pathogenic contaminants [32]. Thus far, very few published works have reported on the simultaneous removal of toxic metal ions, organic dyes, and bacteria. This prompted us to develop an economically and environmentally viable multifunctional nanocomposite material that could effectively eliminate toxic metal ions and coloring dyes as well as inactivate the bacterial growth, henceforth providing safe and unpolluted water. The model contaminants chosen for this study were Pb^2+^, Cu^2+^, CR, and bacteria. Kenaf (*Hibiscus cannabinus*), a natural cellulosic fiber, comprises cellulose, and hemicellulose, and lignin, which comprises functional groups like phenols, amines, and hydroxyls, can be applied as an active site for the adsorption of heavy metals and dyes [33,34]. In our previous study, we successfully demonstrated the application of a kenaf-based nanobiocomposite for the removal of cadmium from an aqueous solution.

This study reports on the development of an economically and environmentally viable multifunctional nanocomposite material that can effectively eliminate toxic metal ions, coloring dyes, and inactivate bacterial growth, providing safe and unpolluted water. The nanocomposite material was developed using AgNPs-modified chitosan and kenaf-based activated carbon for the removal of organic, inorganic, and pathogenic pollutants from water. Batch experiments were administered to assess the adsorption capability, the reaction kinetics, and the implications of critical operating parameters for remediating Pb^2+^, Cu^2+^, and CR. The results demonstrate the effectiveness of the nanocomposite material for the simultaneous removal of heavy metals, dyes, and bacteria from water, providing a promising solution for safe and sustainable water treatment. Additionally, the nanobiocomposite was employed for inhibiting the growth of Gram-positive *Staphylococcus aureus*, and Gram-negative *Escherichia coli* following disk diffusion and minimum inhibitory concentration (MIC) tests.

## 2. Materials and Method

### 2.1. Chemicals

Congo red, lead nitrate (Pb (NO_3_)_2_), copper chloride (CuCl_2_), silver nitrate (AgNO_3_), sodium hydroxide (NaOH), and hydrochloric acid (HCl) were purchased from Sigma–Aldrich (St. Louis, MO, USA). Ken Gro Corp. (Charleston, MS, USA) provided kenaf core fiber. Deionized water (18.2 MΩcm^−1^) was used for conducting all the experiments in the laboratory.

### 2.2. Preparation of Nanobiocomposites

Kenaf core was the raw biomass used for the fabrication of nanobiocomposite. The kenaf-based activated carbon (KAC), was prepared using the facile simple method, which was reported in our previous work [35]. In our present study we prepared two sets of nanobiocomposites by varying the quantity of KAC. An amount of 0.1% of CS was added to 10 mg and 50 mg of KAC, and 1 M of ammonium hydroxide solution, was used to maintain the pH between 6.5 and 7.0. After the addition of 0.01 M silver nitrate (AgNO_3_) drop by drop, the two sets of the mixture were photo irradiated with magnetic stirring by using a spot curing machine (American Ultraviolet Spectrum100 by LESCO). The products of AgNPs-CS-loaded, kenaf-based activated carbon (KAC-CS-AgNPs) were termed as N1 KAC-CS-AgNPs (10 mg KAC) and N2 KAC-CS-AgNPs (50 mg KAC). After the complete reduction of metallic silver to nanoparticles, further centrifugation at 2000 rpm was carried out to isolate the nanobiocomposite from the solution for 5 min. The nanobioadsorbents N1 and N2 KAC-CS-AgNPs were further washed several times (pH~6.5), oven dried for 1–2 h at 40 °C, and stored in a dark container.

### 2.3. Nonaobiocomposite Characterization

A UV-Vis spectrophotometer (PerkinElmer Ltd., Shelton, CT, USA, Model No: Perkin-Elmer Lambda-900) was used for the adsorption studies of the CR dye solutions and to monitor the reduction of the silver (Ag^+^) by the periodic sampling of 1 mL of the aliquots at 20, 30, and 40 min, respectively. The size, structure, and composition of the pristine and spent KAC-CS-AgNPs nanocomposite were performed by scanning electron microscopy (SEM) and energy dispersive analysis X-rays (EDX) using FEI Quanta 200 ESEM. Although we prepared two different sets (N1 and N2) of KAC-CS-AgNPs, only the raw kenaf fiber (KF) and N1 (KAC-CS-AgNPs) were characterized for the BET surface area (S_BET_), total pore volume (V_T_), the average pore diameter (D_avg_), and T plot microarea (T_Area_) by the BET method (Brunauer–Emmett–Teller) using a 3Flex 3500 (Micromeritics Instrument Corp., Norcross, GA, USA).

We used N1 (KAC-CS-AgNPs) for the adsorption of metallic ions and dye. N1 KAC-CS-AgNPs was degassed at 150 °C before N_2_ adsorption for 2 h. The initial and residual concentrations of Cu^2+^ and Pb^2+^ ions were investigated with a graphite furnace atomic absorption spectrophotometer-GAAS (Model No. GFS97 Thermo Scientific Inc., Borken, Germany). The fundamental functional groups present in raw KF, KAC, and N1 KAC-CS-AgNPs were established using a Fourier transform infrared (FTIR) spectrophotometer (Spectrometer Spectrum Two, Perkin Elmer, Inc. Waltham, MA, USA) at a resolution of 4 cm^−1^ and 64 cm^−1^ interferogram scans in the range of 400–4000 cm^−1^ at ambient temperature.

### 2.4. Batch Adsorption Studies

The effects of operational parameters such as contact time, initial concentrations, adsorbent dosage, pH, and temperature on the saturated uptake amount of N1 KAC-CS-AgNPs were studied. To explore the kinetic study for adsorption, the metals and dye solutions with N1 KAC-CS-AgNPs were placed on the shaker at 140–150 rpm for 24 h to ensure an equilibrium was reached at 27 ± 1 °C. To explore the effect of contact time of adsorbate with adsorbent, initial concentrations of Cu^2+^, Pb^2+^, and CR dye, initial pH of the solution, and the adsorbent dosage, the experiments were carried out with 200 mL of aqueous solutions of each of the three adsorbates (Cu^2+^, Pb^2+^, and CR) in an incubator shaker (150 rpm) for different time periods (20–120 min) at 27 ± 1 °C, except the thermodynamical study and the equilibrium were achieved at 120 min. To investigate the kinetics of adsorption, 50 mg of N1 KAC-CS-AgNPs was added to conical flasks containing 200 mL of the heavy metals and dye solutions, and an optimum initial pH was maintained at ~6.5.

To explore the outcome of pH on the adsorption competency of the newly synthesized nanobiocomposite, 25 mg L^−1^ of Pb^2+^, Cu^2+^, and CR solutions were stirred for 120 min with 50 mg of N1 KAC-CS-AgNPs at pH values ranging from 3 to 8. All the batch experiments were performed with the initial concentration of heavy metals and dye at 5 mg L^−1^, 10 mg L^−1^, 15 mg L^−1^, 20 mg L^−1^, and 25 mg L^−1^. After a time interval of 2 h, the samples were withdrawn, centrifuged, and the change in the concentrations of the two metal solutions and CR dye was monitored by GAAS and UV-visible spectrometer at a λ_max_ corresponding to the maximum adsorption for the dye solution (λ_max_ = 497 nm).

The adsorption capacities (mg g^−1^) of Cu^2+^, Pb^2+^, and CR onto nanobiocomposite N1 KAC-CS-AgNPs, at time t ($q_{t}$) and at equilibrium ($q_{e}$), were calculated as follows:(1)$$q_{t}=\frac{(C_{o}-{\;C}_{t})\times \;V}{M}$$
(2)$$q_{e}=\frac{(C_{o}-{\;C}_{e})\times \;V}{M}$$

The removal rate (R%) of the heavy metal ions and dye was calculated using the following relation:(3)$$R\%=\left(\frac{{\;C}_{e}-{\;C}_{t}\;}{C_{e}}\right)\;\times \;100$$
where $C_{o},$ $C_{e},\;\mathrm{and}\;C_{t}\;$denote initial, equilibrium, and residual concentrations (mg L^−1^) of Cu^2+^, Pb^2+^, and CR dye at time 0, 24 h, and t, respectively. V indicates the total volume (L) of metal ions and CR dye solution, and M refers to the mass (g) of nanobiocomposite N1 KAC-CS-AgNPs used for the adsorption study.

### 2.5. Desorption and Reusability Study

The recycling study was performed to explore the economic feasibility of nanobiocomposites (N1). The reusability studies of the KAC-CS-AgNPs were conducted using the optimum experimental parameters (pH~6.5, adsorbent dosage: 0.25 g L^−1^, time of contact: 120 min, metal ions, and dye concentrations: 25 mg L^−1^). Desorption of the analytes from the adsorbent was achieved by washing the collected adsorbent with 1 M sodium hydroxide for ~4 h. In addition, deionized water was used to wash the adsorbent several times before drying it in the oven and reused. The adsorption–desorption process was repeated for 4 cycles.

### 2.6. Antibacterial Test (MIC and MBC)

Investigation of the antibacterial efficacy of the nanobiocomposites was carried out using reference strains of *S. aureus* (ATCC 25923) and *E. coli* (ATCC 25922). We used both N1 and N2 KAC-CS-AgNPs for the antibacterial test. To determine the growth curves of bacterial cells of *E. coli* and *S. aureus* Luria broth were used. Each culture was incubated overnight in a shaking incubator at 37 °C. The minimum inhibitory condition (MIC) and the minimum bactericidal (MBC) were conducted by the standard method to determine the antibacterial efficacy of both N1 and N2 KAC-CS-AgNPs to determine the antibacterial efficacy [36]. MIC is defined as the lowest concentration of AgNPs that inhibited bacterial growth, and MIC value was determined by the standard broth dilution method as described elsewhere [37]. Positive control tubes contained 1 mL of Luria broth and 0.2 mL of *E. coli* and *S. aureus* strains. As a negative control, we used Luria broth. Tubes were incubated at 37 °C for 24 h and each tube was further examined for visual turbidity with the naked eye for the bacterial growth. The MIC endpoint is the lowest concentration of AgNPs where no visible sign of growth was perceived in the test tubes.

After conducting the MIC test, the MBC test was performed and it is defined as the lowest concentration of AgNPs that kills nearly 99.9% of the initial population of pathogens. The tubes that appeared to have no growth or little growth were plated on Luria broth agar plates to comprehend bactericidal or bacteriostatic effects. These plates were incubated at 37 °C for 24 h and then the colonies were noted for the presence of bacterial pathogens.

The disk diffusion test was further conducted to learn the bacterial sensitivity toward both N1 and N2 KAC-CS-AgNPs. The filter disk (~5 mm diameter) was placed on the inoculated agar surface, having uniform bacterial suspensions of *S. aureus* and *E. coli*, and the culture plates were incubated at 37 °C for 24 h. To calculate the diameter of the growth of inhibition or zone of inhibition (ZoI), photographic images of the agar plates were taken.

## 3. Results and Discussion

### 3.1. UV/Vis Spectroscopy

The formation of AgNPs was monitored using a UV/Vis spectrophotometer, and color change was noted after 40 min of photoirradiation of N1 and N2 KAC-CS-AgNPs. The color change in the aqueous solution could be due to the surface excitation of the plasmon resonance phenomenon of silver metal. Therefore, color change can be considered an indication of the successful synthesis of AgNPs. The formation of AgNPs was further analyzed using UV/Vis spectroscopy. Two strong SPR bands, one at 417 nm and the other at 411 nm, were observed for N1 and N2 KAC-CS-AgNPs, as shown in Figure 1 after 40 min of photo-irradiation, thus indicating the formation and successful loading of AgNPs on KAC. Moreover, the broad plasmon band stretched from 330 nm to 800 nm with an absorption tail in the higher wavelength side owing to the multi-size distribution of biosynthesized AgNPs. The findings of the UV/Vis spectra of our material under study exhibited promising resemblance to the optical nature of the reported AgNPs using kenaf seeds [38].

The physicochemical characterizations of KF and N1 KAC-CS-AgNPs are summarized in Table 1. As per the International Union of Pure and Applied Chemistry (IUPAC) system, BET pores are generally classified into three major groups depending on pore diameters: micropores (pore size < 2 °A), mesoporous (pore size 2–50 °A), and macropores (pore size > 50 °A). The average pore diameter of the synthesized N1 KAC-CS-AgNPs was 3.2 nm, and hence it can be classified as a mesoporous medium.

### 3.2. Scanning Electron Microscopy (SEM)

Figure 2 shows the surface morphology of KF, KAC, and N1 KAC-CS-AgNPs, respectively, before the adsorption experiment. KF was dense without visible pores (Figure 2a), indicative of the strong binding of cellulose and hemicellulose with lignin and pectin. The self-activation of KF successfully produced KAC, as judged by the SEM image of KAC (Figure 2b) that exhibited multiple pores with the defined wall surrounding the pores. The presence of multiple pores in KAC can be attributed to the removal of lignin, pectin, and hemicellulose, which enabled the additional stacking space for AgNPs. Figure 2c indicates the successful impregnation of AgNPs onto the KAC-CS matrix. It is evident from Figure 2c that the AgNPs are well distributed and spherical or quasi-spherical in shape, with diameters ranging between 200 and 300 nm. Some clear aggregations are exhibited, which could be due to the concentration of nanobiocomposite upon drying. The elemental characterization of the synthesized AgNPs within the KAC-CS media was also performed using EDX spectroscopy (Figure 2d), which showed strong optical absorption peaks around 3 keV, confirming the chemical identity of AgNPs stabilized within the KAC-CS substrate due to surface plasmon resonance [39]. Figure 2e exhibits the SEM image of the spent N1 KAC-CS-AgNPs, where the visible pores on the surface of N1 KAC-CS-AgNPs disappeared after the adsorption of CR dye, Cu^2+^, and Pb^2+^.

### 3.3. Fourier Transform Infrared Spectroscopy (FTIR)

The FTIR spectra of KF, KAC, and N1 KAC-CS-AgNPs are shown in Figure 3. The FTIR spectrum for raw kenaf fiber (KF) showed a broad peak around 3326 cm^−1^, which can be ascribed to the stretching and bending vibration of O-H groups of phenols, carboxylic acid, and alcohols, as reported in other studies [40]. Moreover, the SEM image of the raw KF further confirmed the findings (Figure 2a). KAC showed an absorption peak at 3670 cm^−1^ that can be assigned to the O–H bond stretching vibration, while those of 2980 cm^−1^ and 2900 cm^−1^ are assigned to asymmetric stretching vibrations of the C–H stretch of alkenes [41]. The peak at 1400 cm^−1^ for KAC is attributed to the alkyl groups of methyl or methylene, as reported by other researchers [42]. The band at 1240 cm^−1^ for KAC is assigned to the C=O stretch for alcohols, carboxylic acids, esters, and ethers, which ascertained the removal of hydrogen and oxygen associated with the raw kenaf fiber [43].

A common strong peak around 1030 cm^−1^ for all KF, KAC, and N1 KAC-CS-AgNPs can be due to C-O-CH_3_ or C-O-C stretching [44]. A weak absorption peak at 880 cm^−1^ was observed for KAC, which is attributed to the bending vibration of the C–H bond of the aromatic ring, which was reported to be like biobased activated carbon [45]. A broad peak appeared at 3360 cm^−1^ for N1 KAC-CS-AgNPs and this could be due to the overlap of N-H stretching vibration of the amino group with -OH stretching vibration, which resulted from the formation of AgNPs on the KAC-CS matrix [46]. A new peak around 1640 cm^−1^ was detected for N1 KAC-CS-AgNPs due to the C-O stretching vibrations of the amide group, which highlighted the interaction of AgNps with the KAC-CS matrix; similar results have been documented [47]. The adsorption peak at 1030 cm^−1^ for N1 KAC-CS-AgNPs can be further attributed to the antisymmetric vibration of C-O-, which resulted from the binding of CS molecule with the nanobiocomposite [48]. Two subsequent signals, around 830 cm^−1^ and 660 cm^−1^, observed for N1 KAC-CS-AgNPs confirmed the existence of out-plane bending of the CH functional group in the nanobiocomposites [49]. Thus, the FTIR results of the KF, KAC, and N1 KAC-CS-AgNPs demonstrated successful loading of AgNPs onto the KAC-CS matrix due to the presence of an ample number of amino groups (NH_2_) derived from CS, which acted as a binding framework.

### 3.4. Batch Adsorption Study

#### 3.4.1. Effect of pH

The pH value of the solution is a crucial parameter that controls the adsorption process due to the ionization of functional groups. The point of zero charge (pH_PZC_) value of the nanobiocomposite N1 KAC-CS-AgNPs was determined to be 6.15 by the method described in the work reported elsewhere [50]. To investigate the effect of pH on the adsorption capacity for Cu^2+^, Pb^2+^, and CR dye on NI KAC-CS-AgNPs, 50 mg of N1 KAC-CS-AgNPs was added to 1 L of the solution containing the known concentration (25 mg L^−1^) of the heavy metals and CR dye. The pH values ranged from 3.0 to 8.0, but the temperature and contact time were kept at 27 ± 1 °C and 120 min, respectively. The pH of the solution was adjusted using 0.1 M HCl and 0.1 M NaOH.

As shown in Figure 4a, the maximum adsorptive removal was observed at pH~7.0 for Cu^2+^and Pb^2+^, and at pH~6.0 for CR dye. The removal rate increased from 60.7% to 68.8% for Cu^2+^, and from 70.6% to 73.4% for Pb^2+^, when the pH was increased from 3.0 to 7.0. A decrease in the removal efficiency for Cu^2+^ to 68.8%, Pb^2+^ to 73.0%, and CR to 74.8% was observed with an increase in the pH from 7.0 to 8.0. We assume that, at low pH (3.0–5.0) values, a high concentration of H^+^ competes with the positive metallic ions and hinders metal ions from approaching the surface of the nanobiocomposites, thus lowering the removal efficiency [51]. Moreover, at pH < pH_PZC_, the surface charge of the adsorbent was positive and the adsorption of metal ions to the surface of the nanobioadsorbent may be obstructed due to the charge repulsion. Conversely, the positively charged metal ions can be easily adsorbed on the negatively charged adsorbent surface [52]. At a higher pH value (pH 8.0), both the Cu^2+^ and Pb^2+^ precipitate in the form of Cu(OH)_2_ and Pb(OH)_2_, respectively, thus retarding the adsorption [53].

Adsorption of CR on N1 KAC-CS-AgNPs increased from 73.5% to 80.0% with increasing pH value from 3 to 6. The maximum adsorptive removal was observed at pH~6.0 (80.0%). The removal capacity diminished with an increase in the pH value from 7 to 8. This occurs due to the protonation of the NH_2_ and OH functional groups and the abundant presence of H^+^ ions on the surface of the adsorbent eventually reducing its dye adsorption capability. Similar pH-dependent results were reported for the adsorption of CR using silver-doped activated carbon [54]. Moreover, based on the FTIR results, the surface of N1 KAC-CS-AgNPs contains oxygen-rich functional groups, i.e., carboxylic groups, carbonyl groups, and phenolic groups. CR dye, being a pH-sensitive anionic dye, has superior activity in an acidic medium compared to a basic medium, as expected. In an acidic solution, the polar acidic groups on CR favor electrostatic interaction with the positive surface charge developed on the surface of N1 KAC-CS-AgNPs. At higher pH (pH > 7), there is a high generation of OH ions that facilitate the breakdown of CR molecules, thereby decreasing the removal efficiency. Similar pH-dependent results were reported for the adsorption of CR on carbon-based silver nanocomposites [55]. Hence, pH 6.5 was chosen for all other adsorption experiments with Cu^2+^, Pb^2+^, and CR dye.

#### 3.4.2. Effect of Contact Time

Contact time is another major factor that influences adsorption. Hence, adsorption experiments were perfomed at the concentration of all three adsorbates at 25 mg L^−1^ and the dosage of N1 KAC-CS-AgNPs at 0.25 g L^−1^, in a time course up to 120 min at pH 6.5 and 27 ± 1 °C. The adsorption equilibrium was assumed to be reached at 120 min as there were no significant differences in the adsorptive removal efficiency of Cu^2+^, Pb^2+^, and CR by N1 KAC-CS-AgNPs at 120 min and 24 h (see the subset in Figure 4b). As shown in Figure 4b, Cu^2+^, Pb^2+^, and CR were rapidly removed within the first 60 min due to the availability of more vacant sites with many active functional groups in N1 KAC-CS-AgNPs. The removal efficiencies of Cu^2+^, Pb^2+^, and CR by N1 KAC-CS-AgNPs after 120 min of contact time were 71.5%, 72.3%, and 75.9%, respectively.

#### 3.4.3. Effect of Temperature

Temperature also influences the adsorption process. Adsorption experiments were performed by setting the reaction temperature at 300 K (27 °C), 310 K (37 °C), 320 K (47 °C), and 330 K (57 °C) while keeping pH at 6.5, the contact time at 120 min, the concentration for all three adsorbates at 25 mg L^−1^, and the dosage of N1 KAC-CS-AgNPs at 0.25 g L^−1^. The adsorption increased from 68.3% to 74.6% for Pb^2+^, 63.5% to 65.1% for Cu^2+^, and 78.9% to 83.9% for CR when the temperature was raised from 300 K to 330 K, as shown in Figure 4c. It is construed that there is an increase in the interparticle diffusion rate in the pores at a higher temperature, thereby enhancing the rate of removal. However, the removal of CR dye decreased, whereas that of Cu^2+^ and Pb^2+^ remained constant, when the temperature was raised to 330 K, as the vacant spaces were filled and the saturation was achieved [56].

#### 3.4.4. Effect of Adsorbent Dosage

The effect of adsorbent dosage on the adsorptive removal of Cu^2+^, Pb^2+^, and CR dye was investigated. Usually, maximum adsorption capacity is seen at low biosorbent dosage due to the complete exposure of diffusion sites, hence we used a low biosorbent dosage for our experiment.

The dosage of N1 (KAC-CS-AgNPs) was varied between 0.125 g L^−1^ and 0.375 g L^−1^ while other experimental parameters were kept the same, i.e., pH at 6.5, the contact time at 120 min, temperature at 27 °C, and the concentration for all three adsorbates at 25 mg L^−1^. 

Figure 4d shows that an increase in N1 dosage from 0.125 g L^−1^ to 0.375 g L^−1^ increased the adsorptive removal of Cu^2+^ from 66.9% to 74.6%, Pb^2+^ from 67.9% to 76.3%, and CR dye from 67.9% to 81.9%. The increase in removal efficiency can be attributed to the presence of more adsorption sites with the increase in adsorbent dosage for the same amount of adsorbate molecules.

#### 3.4.5. Effect of Adsorbate Concentration

Different initial concentrations of metal ions and CR dye, from 5 mg L^−1^ to 25 mg L^−1^, were used to evaluate the effect of adsorbate concentration on adsorption by N1 KAC-CS-AgNPs. Other experimental conditions were kept the same, i.e., pH at 6.5, the contact time at 120 min, temperature at 27 °C (300 K), and the dosage of nanocomposite N1 at 0.25 g L^−1^. As seen in Figure 4e, there was a decrease in the percentage of removal from 89.9% to 71.5% for Cu^2+^, 92.9% to 72.3% for Pb^2+^, and 95.9% to 75.9% for CR dye with an increase in the concentrations of all three adsorbates from 5 mg L^−1^ to 25 mg L^−1^. On the other hand, as the concentration of Cu^2+^, Pb^2+^, and CR dye increased, there was an increase in adsorption capacity (mg g^−1^) from 18.0 to 71.5 for Cu^2+^, 18.6 to 72.3 for Pb^2+^, and 19.2 to 75.9 for CR dye. At low concentrations of Pb^2+^, Cu^2+^, and CR, the availibility of adsorption sites is comparatively high, hence the metallic ions and the reactive dye became attached to the available sites of the nanobiocomposites. As the concentrations of Pb^2+^, Cu^2+^, and CR increased in the solution, there was a decrease in removal efficiency because of the saturation of the availible sites on KAC-CS-AgNPs. It is construed that the increase in the adsorbate concentrations not only escalated the interaction between the adsorbate and the adsorbent, but also provided the required driving force to reduce the resistance to the mass transfer of the adsorbate to the adsorbent [57].

### 3.5. Isotherm Studies

#### 3.5.1. Adsorption Kinetics

The adsorption kinetic depicts the relation of the adsorption capacity with time. To examine the potential rate-controlling factors, the pseudo-first order, the pseudo-second order, and the Elovich models were used to fit the experimental data. The pseudo-first order equation is expressed as follows [58]:(4)$$q_{t}=q_{e}\left(1-e^{-k_{1}t}\right)$$
where $q_{e}$ and $q_{t}$ are the amount of metal ions and dye adsorbed per unit weight of the adsorbent (mg g^−1^) at the equilibrium time and at any time, respectively, and $k_{1}$ is the rate constant of the pseudo-first order model (min^−1^). The linear plot between ln ($q_{e}-q_{t})$ and *t* was used to determine the value of $k_{1}$, as shown in Figure 5a. The correlation coefficients (R^2^) values for Cu^2+^, Pb^2+^, and CR dye were 0.86, 0.91 and 0.94, respectively, for the pseudo-first order kinetic model.

The pseudo-second order equation is expressed as follows:(5)$$\frac{t}{q_{t}}=\frac{1}{k_{2}q_{e}^{2}}+\frac{t}{q_{e}}$$
where $k_{2}$ is the rate constant for the pseudo-second order kinetic model (g mg ^−1^ min^−1^) (Figure 5b), and pseudo-second order parameters $q_{e}$ and $k_{2}$ can be obtained from the slope and intercept of the plot between $\frac{t}{q_{t}}$ and *t*. The R^2^ for the pseudo-second order model for Cu^2+^, Pb^2+^, and CR dye was high (>0.99), indicating that the adsorption with time followed well with the pseudo-second order kinetic mode. It could also be inferred from the high linearity of the plot that chemisorption plays a significant role in the rate-determining step by transferring electrons between the adsorbent and adsorbate, as reported in previous studies [59]. Moreover, the theoretical q_e (cal)_ values were closer to the experimental q_e (exp)_ values (Table 2).

The data obtained from the adsorption experiments were also investigated using the Elovich kinetic model based on Equation (6) [60]:(6)$$q_{t}=\frac{1}{\beta }\;\ln\;(\alpha \beta )+\frac{1}{\beta }\;\ln\;t$$

The Elovich coefficients, *α* (g mg^−1^ min^−1^) and *β* (mg g^−1^), can be obtained from the slope and intercepts of the linear plots between *q_t_* and ln T, as shown in Figure 5c. The high R^2^ (>0.97) obtained by the Elovich kinetic model suggested that the mechanism of chemisorption of Cu^2+^, Pb^2+^, and CR dye was established well with the formation of the heterogeneous surface on N1 KAC-CS-AgNPs.

#### 3.5.2. Adsorption Isotherm

The adsorption isotherm is crucial for the analysis of the mechanism of interaction of the adsorbates with the adsorbent. The Langmuir isotherm model [61] and the Freundlich isotherm model [62] assume homogeneous monolayer adsorption and heterogeneous multilayer adsorption, respectively. The linear form of the Langmuir model is represented by the following equation:(7)$$\frac{C_{e}}{q_{e}}=\frac{1}{bq_{max}}+\frac{1}{q_{max}}\;c_{e}$$
where *c_e_* is the equilibrium concentration of Cu^2+^, Pb^2+^, and CR adsorbed by N1 KAC-CS-AgNPs, *q_e_* is the retention capacity (mg g^−1^) of the metal ions and dye at equilibrium time, *b* is the Langmuir isotherm constant (L mg^−1^) related to the binding energy between the adsorbate and adsorbent, and *q_max_* is the maximum adsorption capacity (mg g^−1^) of the adsorbent.

The linearity was obtained when $\frac{C_{e}}{q_{e}}$ was plotted against *c_e_*, as shown in Figure 6a, where *q_max_* and *b* are calculated as the slope and the intercept, respectively. The *q*_*max*_ (mg g^−1^) was found to be 79.37, 78.13, and 80.65 for Cu^2+^, Pb^2+^, and CR, respectively. The R^2^ was found in the order of CR (0.99) > Pb^2+^ (0.95) > Cu^2+^ (0.93). In general, the nature of the adsorption process is determined from the value of R_L_ (separation factor) [63]:$$R_{L}=\frac{1}{{\left(1+bc_{0}\right)}_{\;}}$$
where $c_{0}$ (mg L^−1^) is the initial metal ions and dye concentration and *b* (L mg^−1^) is the Langmuir constant related to the energy of adsorption. The value of *R_L_* lies between 0 and 1 when the adsorption process is favorable, whereas it is greater than 1 when unfavorable. Also, it indicates whether the adsorption process is linear (*R_L_* = 1) or irreversible (*R_L_* = 0). The calculated *R_L_* values for the present investigation were Cu^2+^ = 0.0492, Pb^2+^ = 0.0393, and CR = 0.0238, indicating that the adsorption process of metals and dye onto N1 KAC-CS-AgNPs was favorable and more likely to be irreversible (Table 2).

The linear form of the Freundlich model is given by the following equation:(8)$$\ln\;q_{e}=\ln\;K_{f}+\frac{1}{n}\;\ln\;c_{e}$$
where *K_f_* is the Freundlich constant and *n* is the Freundlich exponent. Both can be determined from the plot between ln *q_e_* and ln *c_e_*, as depicted in Figure 6b. The isotherm parameters for the adsorption of Cu^2+^, Pb^2+^, and CR are given in Table 2.

The R^2^ values were found to be as high as 0.99 for Pb^2+^, Cu^2+^, and CR. Thus, Pb^2+^ and Cu^2+^ adsorption fitted better to the Freundlich isotherm model than the Langmuir isotherm model, whereas both isotherm models fitted well the CR adsorption. The value of 1/n obtained from the Freundlich isotherm model was smaller than 1, which represented favorable adsorption conditions and adsorbate molecules Pb^2+^, Cu^2+^, and CR were adsorbed on the heterogeneous surface [64]. By comparing the values from Table 2, it was evident that the Freundlich Isotherm model provided the best fit with experimental data.

#### 3.5.3. Thermodynamic Studies

Thermodynamic studies are crucial in establishing the spontaneity and feasibility of adsorption processes. Thermodynamic data were determined using Equations (9)–(11) [65]:(9)ΔG=ΔH−TΔS
(10)$$K_{c}=\frac{q_{e}}{c_{e}}$$
(11)$$lnK_{c}=\frac{\Delta S\;}{R}-\frac{\Delta H}{RT}$$
where ∆*G*^0^ is the Gibbs free energy change, ∆*H*^0^ is the change in enthalpy, and ∆*S*^0^ is the change in entropy. *K_c_*, *R*, and *T* are the equilibrium constant, the gas constant (8.314 J mol^−1^ K^−1^), and the absolute temperature (*K*), respectively.

The thermodynamical values of ∆*H*^0^, ∆*S*^0^, and ∆*G*^0^, as well as other calculated parameters, are given in Table 3. The negative values of Δ*G*^0^ validated that the adsorption process was spontaneous. Furthermore, the decrease in the values of Δ*G*^0^ with the increment of the temperature indicated that the adsorption process was more spontaneous at elevated temperatures. Generally, it has been noted that, for physisorption, the change in free energy is between (−)50 kJ mol^−1^ and 0 kJ mol^−1^, and the values of Δ*G*^0^ obtained in this study were within this range. Therefore, physisorption could have also occurred along with chemosorption in the adsorption process [66]. In the adsorption study, ∆*H*^0^ gives an insight into the nature and mechanism of adsorption processes. The positive values of ∆*H*^0^ (J mol^−1^) for Pb^2+^ (10,369), Cu^2+^ (3752), and CR (1620) suggested an endothermic reaction. The positive values of ∆*S*^0^ (J mol^−1^) for Pb^2+^ (67.2), Cu^2+^ (31.9), and CR (26.9) suggested an increased randomness, leading to the greater affinity of the adsorbent toward the adsorbates. These results agreed with earlier findings [67].

### 3.6. Reusability

Sustainable remediation demands the easy regeneration of the spent adsorbents after the uptake of pollutants, but it maintains its adsorption capacity after several cycles (i.e., reusability). To evaluate the recyclability of the N1 KAC-CS-AgNPs, the adsorbates concentration at 25 mg L^−1^ and the N1 dosage at 0.25 g L^−1^ were run for adsorption for 120 min at pH 6.5 and 27 °C. It was found that the adsorptive removal by N1 KAC-CS-AgNPs was almost constant for three cycles, resulting in 75%, 72%, and 71% for CR, Pb^2+^, and Cu^2+^, respectively. It slightly decreased by ~2% in the fourth cycle (Figure 7). Thus, the superior reusability and the applicability of N1 KAC-CS-AgNPs for remediating multiple pollutants were demonstrated.

### 3.7. Antibacterial Efficacy

The antimicrobial performance was evaluated for both N1 and N2 KAC-CS-AgNPs by the disc paper diffusion method using *E. coli* and *S. aureus*. The controls were 0.05 M silver nitrate solution (AgNO_3_) and Chitosan-based AgNPs (CS-AgNPs). Inhibition zones for *E. coli* of ~12 mm and, for *S. aureus*, around ~10 mm were determined. Based on the diameter of the inhibition zone obtained, the higher effectiveness of N1 KAC-CS-AgNPs; N2 KAC-CS-AgNPs was detected, then CS-AgNPs, and finally AgNO_3_ (0.05 M). The precise mechanism of AgNPs on bacterial inactivation is still under research. Various studies have reported that AgNPs first attaches to the pathogenic cell membrane and passes through cell wall and cytoplasmic membrane. The cell wall is disrupted by the generation of reactive oxygen species (ROS) by the AgNPs and the NPs react with the sulfur and phosphorus of the DNA molecules of the cell, thus damaging the entire cell organelles and causing the death of the cell [68].

The bacteriostatic and bactericidal effects of synthesized nanobiocomposites N1 and N2 were screened by the determination of MIC and MBC against *E. coli* and *S. aureus* (SA). Figure 8 and Table 4 show the typical results carried out for the bactericidal test of N1 nanobiocomposite toward *E. coli* and *S. aureus*. The cell growth was quantified by counting colonies, and no growth indicated a bactericidal effect. Nanobiocomposites (KAC-CS-AgNPs) showed a strong antibacterial activity, with MIC concentration of 32 µg/mL for *E. coli* and 43.6 µg/mL for *S. aureus*, and the MBC concentrations of 44 µg/mL and 51.1 µg/mL for *E. coli* and *S. aureus* were calculated.

### 3.8. Comparison Study

Several studies were conducted on the adsorption of metal ions and dye and on the bacterial disinfection with the different types of nanoadsorbents. In Table 5, the adsorption capacity of KAC-CS-AgNPs is compared with other biobased activated carbon with silver-doped nanocomposites for the remediation of metals ions, dye, and bacteria. The comparative evaluation of the adsorption capacities of various types of low-cost biobased silver nanoadsorbents, for the removal of metals and organic dye and inactivation of pathogens, clearly indicated that the newly fabricated (KAC-CS-AgNPs) nanobiocomposite is an excellent nanobioadsorbent for removing multiple pollutants from wastewater.

## 4. Conclusions

The current research presents a novel, multifunctional benign nanobiocomposite fabricated via a fast, facile, and green method, by loading AgNPs in kenaf-based activated carbon in the presence of chitosan as a stabilizing agent. The fabricated nanobiocomposite KAC-CS-AgNPs exhibited a superior remediation of toxic pollutants, including Cu^2+^, Pb^2+^, and carcinogenic CR dye. The results further indicated that adsorption kinetics, as well as the adsorption isotherm, were well described by the pseudo-second order kinetic model and Freundlich and Langmuir models. Furthermore, the thermodynamic studies revealed that the adsorption process was spontaneous, feasible, and endothermic with chemical interactions playing a dominant role. KAC-CS-AgNPs also showed notable reusability with a minimal decrease in its adsorption efficiency after four successive adsorption cycles. Additionally, KAC-CS-AgNPs showed bacterial resistance towards Gram-positive and Gram-negative bacteria. Overall, the KAC-CS-AgNPs is a promising candidate for the efficient adsorption of toxic heavy metal ions and industrial dye, as well as for superior bacterial disinfection.

## Figures and Tables

**Figure 1 biomolecules-13-01054-f001:**
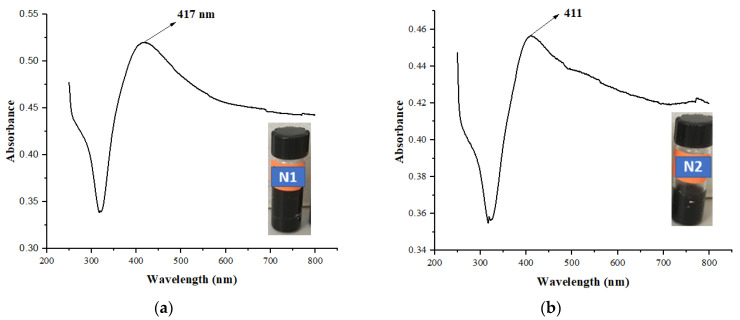
UV/Vis absorption spectra of nanobiocomposites N1 KAC-CS-AgNPs (**a**) and N2 KAC-CS-AgNPs (**b**).

**Figure 2 biomolecules-13-01054-f002:**
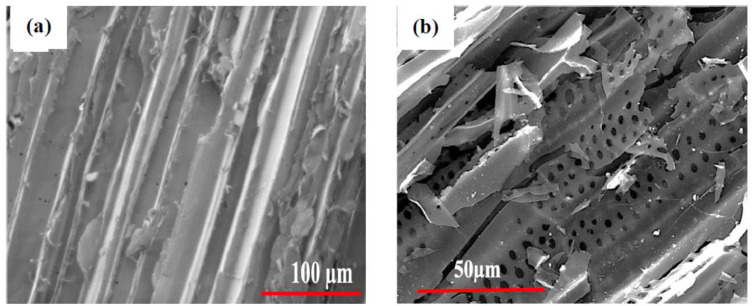
SEM image of kenaf Fiber (KF) at ×1000—(**a**), SEM image of KAC at ×2000—(**b**), SEM image of N1 KAC-CS-AgNPs at ×5000 before adsorption study—(**c**), EDX spectrum of N1 KAC-CS-AgNPs before adsorption study—(**d**), and SEM image of spent KAC-CS-AgNPs at ×5000 after adsorption study with Cu^2+^, Pb^2+^, and CR—(**e**).

**Figure 3 biomolecules-13-01054-f003:**
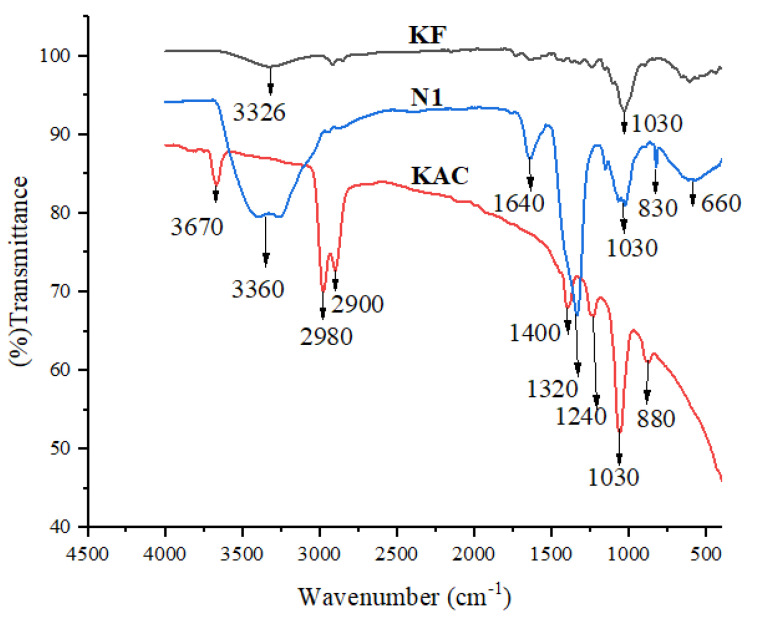
FTIR spectra for KF, KAC, and N1 KAC-CS-AgNPs.

**Figure 4 biomolecules-13-01054-f004:**
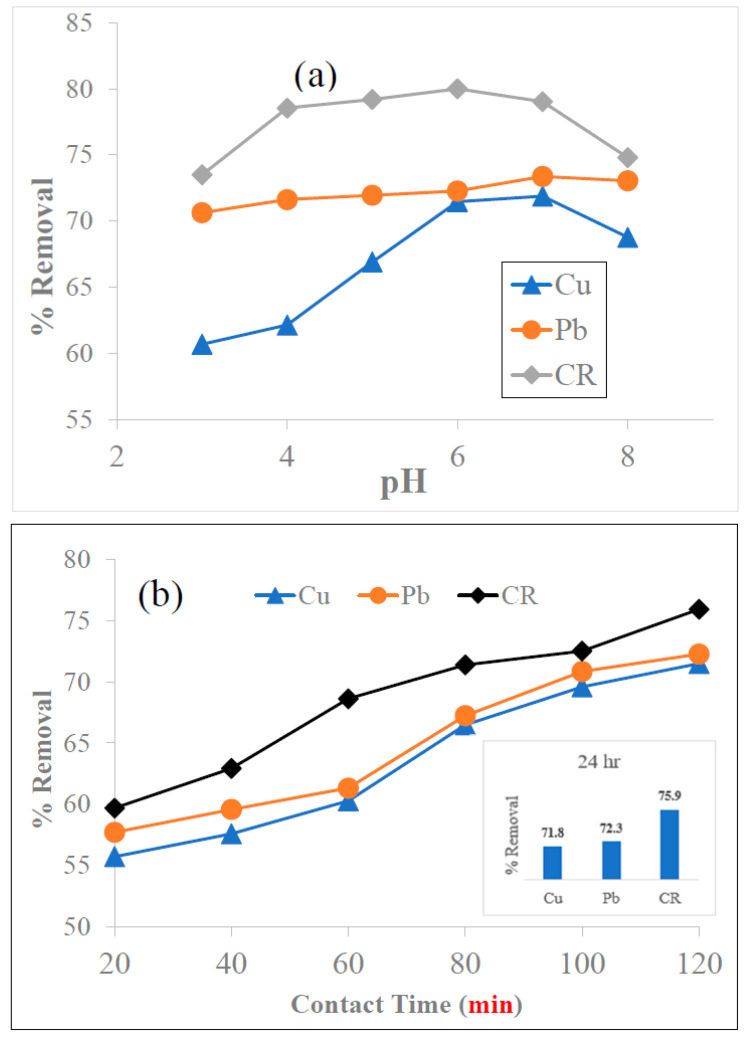
Effect of (**a**) pH, (**b**) contact time, (**c**) temperature, (**d**) adsorbent dosage, and (**e**) initial metal ions and dye concentration on the adsorptive removal of Cu^2+^, Pb^2+^, and CR dye by N1 KAC-CS-AgNPs.

**Figure 5 biomolecules-13-01054-f005:**
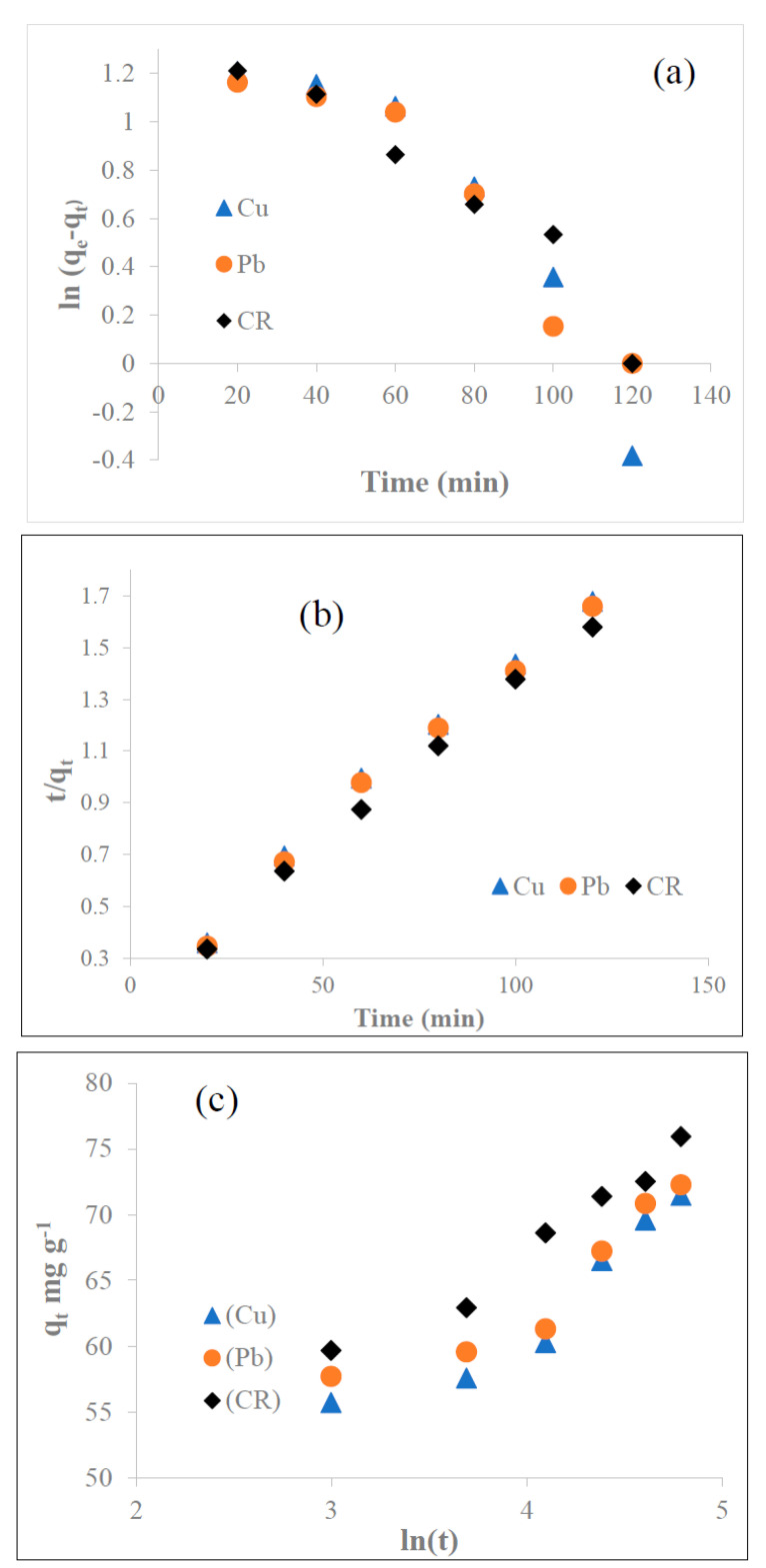
Kinetic models of pseudo first-order (**a**), pseudo second-order (**b**), and Elovich (**c**) for the adsorption of 25 mg L^−1^ of Cu^2+^, Pb^2+^, and CR at pH 6.5 and 27 °C.

**Figure 6 biomolecules-13-01054-f006:**
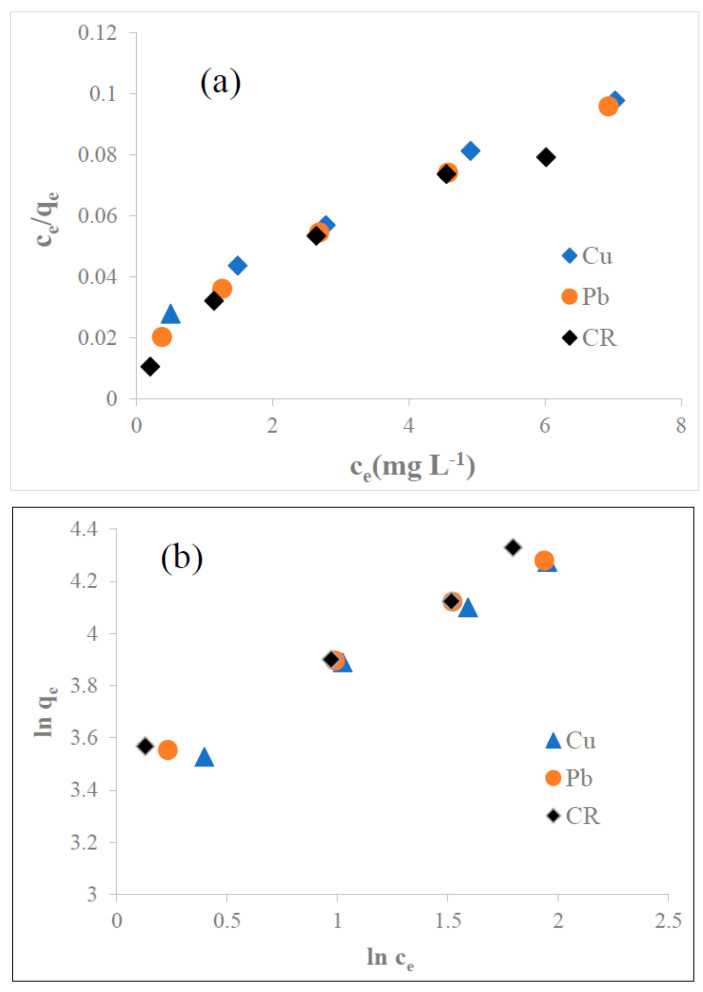
Adsorption isotherm models, (**a**) Langmuir and (**b**) Freundlich, of Cu^2+^, Pb^2+^, and CR at 25 mg L^−1^ onto N1 KAC-CS-AgNPs at 0.25 g L^−1^, pH 6.5, and 27 °C (300 K).

**Figure 7 biomolecules-13-01054-f007:**
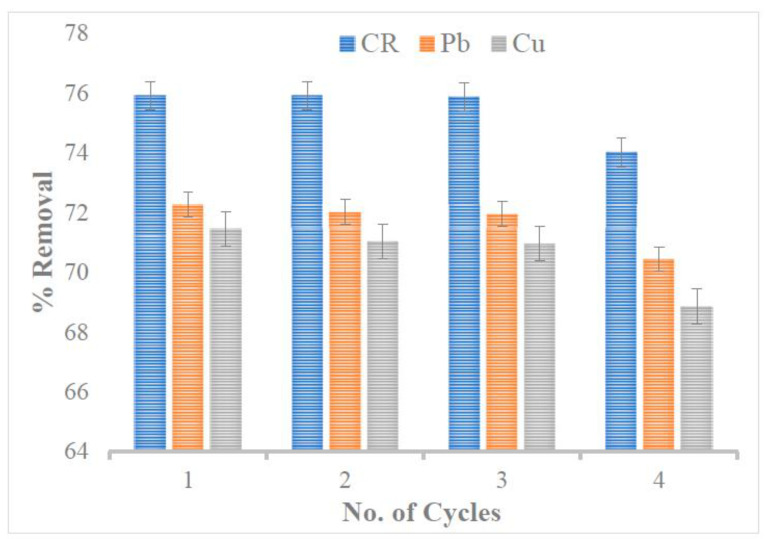
Reusability test for N1 KAC-CS-AgNPs up to four cycles.

**Figure 8 biomolecules-13-01054-f008:**
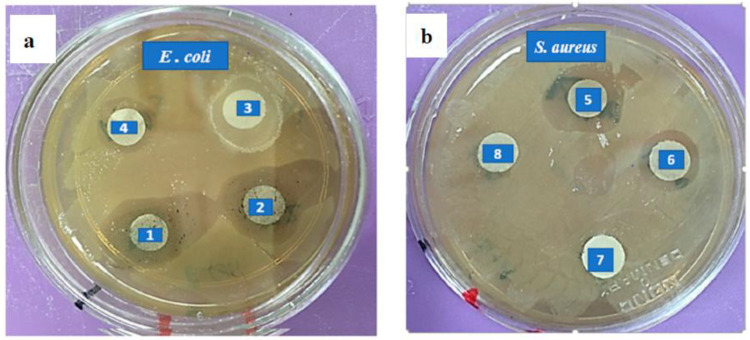
Zone of inhibitions against *E. coli* (**a**) and *S. aureus* (**b**) after 24 h of incubation at 37 °C. Numbers 1 and 5: N1 KAC-CS-AgNPs, 2 and 6: N2 KAC-CS-AgNPs, 3 and 7: CS-AgNPs, and 4 and 8: AgNO_3_ (0.05 M).

**Table 1 biomolecules-13-01054-t001:** BET analysis of raw kenaf biomass (KF) and nanobiocomposite N1 KAC-CS-AgNPs.

Sample	S_BET_ (m^2^ g^−1^)	V_T_ (cm^3^ g^−1^)	D_Avg_ (nm)	T _Area_ (m^2^ g^−1^)
KF	3.2	0.000	5.50	0.00
N1 KAC-CS-AgNPs	204.4	0.003	3.20	8.30

S_BET_—BET surface area, V_T_—total pore volume, D_Avg_—average pore diameter, T_Area_—T plot micro area.

**Table 2 biomolecules-13-01054-t002:** Isotherms and kinetic constants and correlation coefficients.

Model		Parameter	Cu^2+^	Pb^2+^	CR Dye
Isotherm	Langmuir	q_max_ (mg g^−1^)	79.37	78.13	80.65
		q_exp_ (mg g^−1^)	71.88	72.27	75.93
		b (L mg^−1^)	0.773	0.977	1.638
		R^2^	0.928	0.950	0.997
		R_L_	0.049	0.039	0.024
	Freundlich	K_f_	0.0387	0.0343	0.0267
		1/n	0.524	0.469	0.394
		R^2^	0.993	0.996	0.992
Kinetic	Pseudo-first order	$k_{1}$ (min^−1^)	−0.0153	−0.0129	−0.0114
		R^2^	0.863	0.904	0.940
	Pseudo-second order	K_2_	0.00058	0.00116	0.00131
		q_e_ (cal) mg g^−1^	72.46	77.52	80.65
		q_e_ (exp) mg g^−1^	71.88	72.27	75.94
		R^2^	0.990	0.993	0.997
	Elovich	α (mg g^−1^ min^−1^)	17.939	28.954	34.914
		β (g mg^−1^)	0.0680	0.0672	0.0636
		R^2^	0.973	0.967	0.972

**Table 3 biomolecules-13-01054-t003:** Thermodynamic coefficients for the adsorption of Cu^2+^, Pb^2+^, and CR by N1 KAC-CS-AgNPs.

Adsorb-Ate	ΔS (kJ mol^−1^)	ΔH (kJ mol^−1^)	R^2^	ΔG (kJ mol^−1^)			
				300 K	310 K	320 K	330 K
Cu^2+^	31.9	3752	0.968	−5744.3	−6176.3	−6518.2	−6692.2
Pb^2+^	67.2	10,369	0.978	−9716.8	−10,514.1	−11,310.2	−11,692.4
CR dye	26.9	1620	0.818	−6320.8	−6814.7	−7161.7	−7101.4

**Table 4 biomolecules-13-01054-t004:** Zone of inhibition using disc diffusion test against *E. coli* and *S. aureus* using kenaf-based silver nanocomposites, chitosan-based silver nanocomposites, and silver nitrate.

Disinfectant	Diameter (mm) of Zone of Inhibition (ZoI)	
	*E. coli*	*S. aureus*
N1 KAC-CS-AgNPs	12.0	10.0
N2 KAC-CS-AgNPs	12.0	9.0
CS-AgNPs	8.0	7.0
AgNO_3_ (0.05 M)	6.0	5.0

**Table 5 biomolecules-13-01054-t005:** Reported adsorption capacity of metal ions and dye and microbes by silver-nanoparticles-loaded carbon nanocomposites.

Adsorbent	Dosage (g L^−1^)	Removal of Dyes and Metals (% Removal or Adsorption Capacity)			Microbial Pathogens (Zone of Inhibition)				Refs.
Jute-ACNPs	0.28	NA	NA	NA	*E. coli*- 20 mm	*P. aeruginosa* 8 mm	*B. subtilis* 25 mm	*S. aureus-* 23 mm	[69]
Powder AC-AgNPs (Commercial)	NA	NA	NA	NA	*E. coli*-18 mm	*P. aeruginosa* NA	*B. subtilis* NA	*NA	[70]
Azadirachta indica (Neem)-ACAg/ZnO	0.4	NA	NA	NA	*E. coli*- 6 mm	*P. aeruginosa* 6 mm	*B. subtilis* NA	*S. aureus-*18 mm	[71]
AC-AgNPs	0.1	NA	NA	NA	*E. coli* NA	*P. aeruginosa* NA	*B. subtilis* NA	NA	[72]
Hildegardia barteri-AC-NPs	1	Congo red (q_exp_) 161.29 mg g^−1^	NA	NA	*E. coli* NA	*P. aeruginosa* NA	*B. subtilis* NA	*S. aureus* NA	[73]
Ag NPs-MWCNT	0.5	NA	NA	Copper (q_exp_) 58.02 mg g^−1^	*E. coli* NA	*P. aeruginosa* NA	*B. subtilis* NA	*S. aureus* NA	[74]
Magnetite @AC-AgNPs	0.5	NA	Lead (q_exp_) 75 mg g^−1^	NA	*E. coli* 28 mm	*P. aeruginosa* NA	*B. subtilis* 25 mm	*S. aureus* NA	[75]
Ag-NPs-AC	0.5	Congo red (q_exp_) 66.7 mg g^−1^	NA	NA	*E. coli * NA	*P. aeruginosa* NA	*B. subtilis* NA	*S. aureus* NA	[49]
Castor seed-AC-AgNPs	2.0	NA	NA	NA	*E. coli* 9.3 mm	*P. aeruginosa* NA	*B. subtilis* NA	*S. aureus* NA	[76]
AC-Ag-SiO_2_ nanocomposite	1.0	NA	Lead (q_exp_) 81.3 mg g^−1^	Copper (q_exp_) 84.8 mg g^−1^	*E. coli* NA	*P. aeruginosa* NA	*B. subtilis* NA	*S. aureus* NA	[77]
KAC-CS-AgNPs	0.25	Congo red (q_exp_) 75.9 mg g^−1^	Lead (q_exp_) 72.3 mg g^−1^	Copper (q_exp_) 71.5 mg g^−1^	*E. coli*-12 mm	NA	NA	*S. aureus-*10 mm	**This work**

*NA—Not available.

## Data Availability

The data presented in this study are available on request from corresponding authors. The data are not publily available due to privacy restrictions.

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
