# Peer review of "Bioinspired Synthesis of Silver Nanoparticles for the Remediation of Toxic Pollutants and Enhanced Antibacterial Activity"

_biomolecules, 2023, doi:10.3390/biom13071054_

Round 1

Reviewer 1 Report

1. In the Introduction (lines 82-84) auhors mention that in a previous work Cd adsorption was studied. Please cite the reference. In addition, it would be interesting to have Cd adsorption studies in this work.

2. In line 263, when discussing FTIR, authors mention "The band at 1,240 cm-1 for KAC 262 is assigned to the C=O stretch for...". That band is assigned to C-O and not C=O. Please correct.

3. Figure 5c makes no sense being presented with the abciss axis starting at "0". Please revise it starting at a value of "2".

4. Authors show the results for the reusability of the composite mateirla in the removal of the selected metal ions and dye. What is missing in the Experimental section is the descirption of the protocol for the reusability of the adsorbent (i.e. how the adsorbent is regenerated). This must be added.

5. In the performance comparison of the different materilas (Table 5) authors compare their achievements with other results found in the literature. This is welcome, but it is defying to find a rational as most of the chosen benchmarking materials were used for different metals thna those studied here. Please try to reassess these benchmarking literature studies to mirror more similar ion removal and dye.

6. In additon, in the Conclusions, authors state that "KAC-CS-AgNPs showed a strong inhibitive activity towards Gram-547 positive and Gram-negative bacteria." Although inhibitory activity was found, for the case of S. aureus the reported activity was the lowest (Table 5) and that can be misleading. Rephrase this.

7. There are several formatting issues that must be revised throughout the manuscript.

Therefore, major revision is recommended.

Reviewer 2 Report

Sujata Mandal et al. synthesized Ag nanoparticles for water remediation. The work is well-planned and executed. However, there are minor issues with the consistency of fonts and representations. Please fix these issues in the revised version.

1.       Please pay attention to font sizes and styles.

2.       The font styles on line 122 are not consistent. This pattern can be seen throughout the entire manuscript. Please fix this in the revised version.

3.       Cite a few relevant recent biomaterials-based water remediation works. Example:

https://doi.org/10.1016/j.heliyon.2023.e16600; https://doi.org/10.1016/j.seppur.2022.122935;

https://doi.org/10.3390/nano13030399

4.       Authors have used 'hours' and 'h' in a few instances. Maintain consistency in these words. Similarly, instead of minutes, use the term 'min'.

5.       The authors reported in section 3.4.4 that increasing the dosage led to an increase in the adsorptive capacity. What will happen to absorption capacity if a dosage of more than 375 g/L is used?

6.       Authors should explain and justify the findings under section 3.4.5, "Effect of Adsorbate Concentration," in their works.
